# Bacterial detoxification of plant defence secondary metabolites mediates the interaction between a shrub and frugivorous birds

Beny Trabelcy [1] ✉, Nimrod Shteindel [1], Maya Lalzar[2], Ido Izhaki [1] & Yoram Gerchman [1,3] ✉

Many plants produce fleshy fruits, attracting fruit-eating animals that disperse the seeds in their droppings. Such seed dispersal results in a conflict between the plant and the animal, as digestion of seeds can be highly beneficial to the animal but reduces plant fitness. The plant *Ochradenus baccatus* uses the myrosinase-glucosinolates system to protect its seeds. We show that hydrolysis of the *O. baccatus* fruit glucosinolates by the myrosinase enzyme inhibited digestive enzymes and hampered digestion in naïve individuals of the bird *Pycnonotus xanthopygos*. However, digestion in birds regularly feeding on *O. baccatus* fruits was unaffected. We find that *Pantoea* bacteria, dominating the gut of these experienced birds as well as the fruits, thrive on glucosinolates hydrolysis products in culture. Augmentation of *Pantoea* protects both naïve birds and plant seedlings from the effects of glucosinolates hydrolysis products. Our findings demonstrate a tripartite interaction, where the plant-bird mutually beneficial interactions are mediated by a communal bacterial tenant.

Seed dispersal is a fundamental life cycle stage in plants. Seed dispersal functions in the reduction of competition for resources between parent and offspring and among offspring[1], and enables recruitment into new habitats[2]. Endozoochory is a common seed dispersal strategy involving the production of fruits to attract frugivores that feed on the fruits and disperse seeds in their droppings[3]. Beside dispersal, endozoochory enhances germination by separating seeds from the pulp (often containing germination inhibitors), fertilizing the ground with fecal material, and scarification of the seed coat[4–7]. Interaction between plants and frugivores is assumed mutualistic, with the consumer benefiting from nutrients in fruits and the plant benefiting from seed dispersal[8,9]. Nevertheless, this relationship can be complex. Seeds are rich in proteins generally lacking in the fruit pulp[10,11], making seed digestion beneficial[12] and thus hampering plant fitness[13,14], resulting an evolutionary arms race between plants and frugivores[15]. Hence, plants face a delicate balance—attracting seed dispersers while evading seed predation and digestion.

Secondary metabolites (SMs) were suggested to play a role in this trade-off[16,17]. Glucosinolates (GSLs) are common SMs in plants of the Brassicales order. Intact GSLs are harmless, but when mixed with the myrosinase enzyme (MYR; EC 3.2.1.147), GSLs go through rapid hydrolysis, releasing toxic products (aka the "mustard oil bomb")[18], which protect Brassicales plants from herbivory[18,19] and reduce intake of plant tissue by herbivores[20]. To prevent self-inflicted damage to the plant, the GSLs and MYR are separately compartmentalized, mixed only upon damage to the tissue[21,22].

The desert plant *Ochradenus baccatus* (Delile 1813 [Resedaceae]) produces sweet fleshy fruits[23], attracting a variety of animals[24]. The fruit pulp mainly contains water and sugar, whereas the seeds are rich in protein, fat, and starch[25], making them a desired food. Previously, we showed that *O. baccatus* uses the MYR-GSL mechanism to reduce seed predation by the spiny mouse *Acomys russatus* (an effective seed predator)[23], by compartmentalizing the GSLs in its pulp and the MYR in its seeds[23]. Animals that consume and digest the seeds with the pulp mix the two and thus activate the MYR-GSL mechanism. As a result, the animals are exposed to the GSLs hydrolysis product.

In the desert, *O. baccatus* fruits are frequently eaten by *Pycnonotus xanthopygos* (Hemprich and Ehrenberg 1833; white-spectacled

[1]Department of Evolutionary and Environmental Biology, Faculty of Natural Sciences, University of Haifa, Haifa 3498838, Israel. [2]Bioinformatic Unit, University of Haifa, Haifa, Israel. [3]Oranim College, Kiryat Tivon 3600600, Israel. ✉ e-mail: benytrabelsy@gmail.com; gerchman@research.haifa.ac.il

bulbuls)[24,26], a songbird common throughout Israel. *Pycnonotus xanthopygos* swallow the whole fruits and excrete seeds[26] and are considered a major seed disperser of *O. baccatus*. Although passage through the digestive system of *P. xanthopygos* significantly increases the germination of seeds, the birds digest up to 80% of the consumed seeds, making the interaction multi-faceted[15]. Consumption of *O. baccatus* fruits also results in the reduced digestive ability of the birds[15], suggesting a plant defense mechanism against seed predation by the *P. xanthopygos*.

Here we report that the MYR-GSL hydrolysis products of *O. baccatus* fruits inhibit digestive enzymes, hamper *P. xanthopygos* food digestibility, and have allelopathic effects on seedling growth of *O. baccatus*. Birds regularly feeding on *O. baccatus* fruits acquire fruit-associated bacteria that detoxify the MYR-GSL hydrolysis products, recovering digestibility. Finally, the same bacteria passed through the bird's digestive system improved seedling growth and prevented the MYR-GSL hydrolysis products effect. These results demonstrate a tripartite, multi-kingdom interaction mediated by bacteria and affecting both plant and bird phenotypes.

## Results

### *Ochraceous baccatus* fruits hamper digestion in naïve *P. xanthopygos*

To test the effect of *O. baccatus* fruits on bird digestion, we captured birds from two locations 95 km apart (Fig. 1a). Birds were captured on *O. baccatus* plants (hereafter experienced birds) in one (Southern) location, and another group of birds was captured at a (Northern) location far from the edge of *O. baccatus* distribution (hereafter naïve birds) and acclimated in captivity. We assume that birds collected outside *O. baccatus* distribution region (naïve birds) had no previous encounter with that plant. Naïve birds acclimated to banana (Fig. 1b, W4–6, yellow) showed reduced digestive ability when fed banana + *O. baccatus* pulp + seeds extract (Fig. 1c, B+P+S) as compared to experienced birds acclimated to *O. baccatus* fruits (Fig. 1b, W4–6, blue and Fig. 1c, B+P+S). Feeding on banana or banana mixed with either *O.*

*baccatus* pulp or seed extract did not have such an effect, regardless of the acclimation diet (Fig. 1c: B, B+P and B+S). These results demonstrate that consumption and thus mixing of *O. baccatus* fruit pulp and seeds hinders digestion in vivo and that birds experienced with *O. baccatus* may acquire resistance to the digestibility-reducing effects of MYR-GSL hydrolysis products.

### Characterization of the MYR-GSL defense mechanism in *O. baccatus* fruits and its effect on bird digestion

HPLC analysis of the *O. baccatus* pulp identified the presence of BenzylGSL (Glucotropaeolin) as the main glucosinolate in the fruit pulp (Fig. 2a). Mixing *O. baccatus* fruit pulp and seeds resulted in the appearance of Benzyl isothiocyanate (BITC), as identified by GC-MS (Fig. 2b). BITC formation is due to the activity of the MYR enzyme found in the seeds that hydrolyzes intact BenzylGSL found in the fruit pulp (Fig. 2c). To further explore this interplay in an in vivo context, we used HPLC to quantify the BenzylGSL concentration after the passage of whole fruits through the bird digestive tract (Fig. 2d). BenzylGSL decomposed rapidly in the bird digestive system—starting from $14.38 \pm 1.37 \, \mu g/g$ fresh weight (FW) in the *O. baccatus* fruits to $1.54 \pm 0.16 \, \mu g/g$ FW in naïve birds and $1.77 \pm 0.2$ in experienced bird droppings (Fig. 2d). The 88–89% decrease in concentration is likely due to the hydrolysis of the BenzylGSL in the bird digestive system. No BITC was detected in the droppings of birds feeding on *O. baccatus* fruits, probably because BITC readily binds to proteins[27], such as in the digestive system of the birds, creating conjugates that will be undetectable in the GC/MS.

To test if the reduction in the digestibility of birds feeding on seed extract mixed with fruit pulp juice is mediated by the BenzylGSL hydrolysis products, we tested the effect of BITC on the activity of two major digestive enzymes, amylase and lipase, both likely to take part in seed digestion[15] (commercial enzymes were used; see Methods section). Reduced activity of both enzymes was evident (Fig. 2e, f), suggesting that the BenzylGSL hydrolysis products inhibit digestive enzymes and hamper digestion, highlighting the biochemical

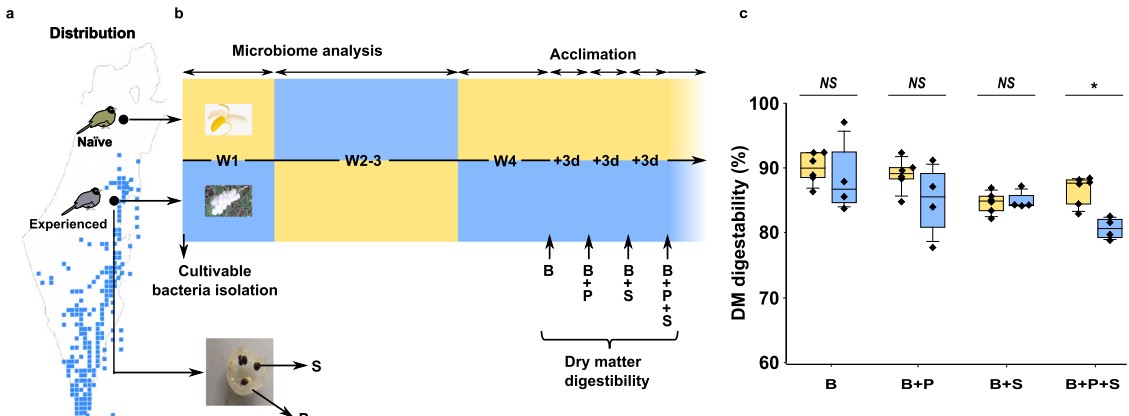

**Fig. 1 | *Ochradenus baccatus* distribution map and *Pycnonotus xanthopygos* collection sites; experimental timeline; mixture of *O. baccatus* seed extract and fruit pulp inhibits digestion in naïve birds. a** *Ochradenus baccatus* distribution in Israel (blue dots designate sightings) and birds capture sites (black dots). Fruits were collected near the northern coast of the Dead Sea, Israel. Birds were captured near *O. baccatus* (i.e., experienced birds; n = 6) or outside its distribution (i.e., naïve birds; n = 4). Aerial distance between the two habitats is 98 km, with the longest daily flight distance reported for bulbuls being 5 km[26]. **b** Experimental timeline (W week, d days): In week 1, recently captured naïve birds fed on bananas (yellow background) and recently captured experienced birds fed on *O. baccatus* fruits (blue background). Beginning week 2, birds' diets were switched (W2–3), and switched again in week 4. Droppings for culturable bacteria were collected on the first day of W1 and for microbiome analysis daily through weeks

1–3. For digestibility experiments, birds were acclimated for 1 week (W4) on *O. baccatus* fruits or bananas. *Ochradenus baccatus* seeds (S) and pulp (P) were manually separated, seeds ground and extracted in water and pulp squeezed for juice, both sterilized by 0.22 μm filtration. Once every 3 days (Figs. 1b and 3d), birds were fed for 30 min with a designated diet (banana + water, B; banana + *O. baccatus* pulp juice, B+P; banana + *O. baccatus* seed extract, B+S; or banana mixed with both *O. baccatus* pulp and seed extract, B+P+S) and digestibility calculated. Between experiments, birds fed on bananas or *O. baccatus* (Fig. 1b, yellow and blue background, respectively). **c** Dry matter (DM) digestibility (%) by birds fed different diets. Box centerlines are median, box limits are lower (Q1) and upper quartile (Q3), and whiskers are 5–95 percentiles; two-tailed *t*-test for two independent samples, $t_8 = 4.306$, $P = 0.003$, n = 4 for naïve birds and 6 for experienced birds. NS not significant. Source data are provided as a Source Data file.

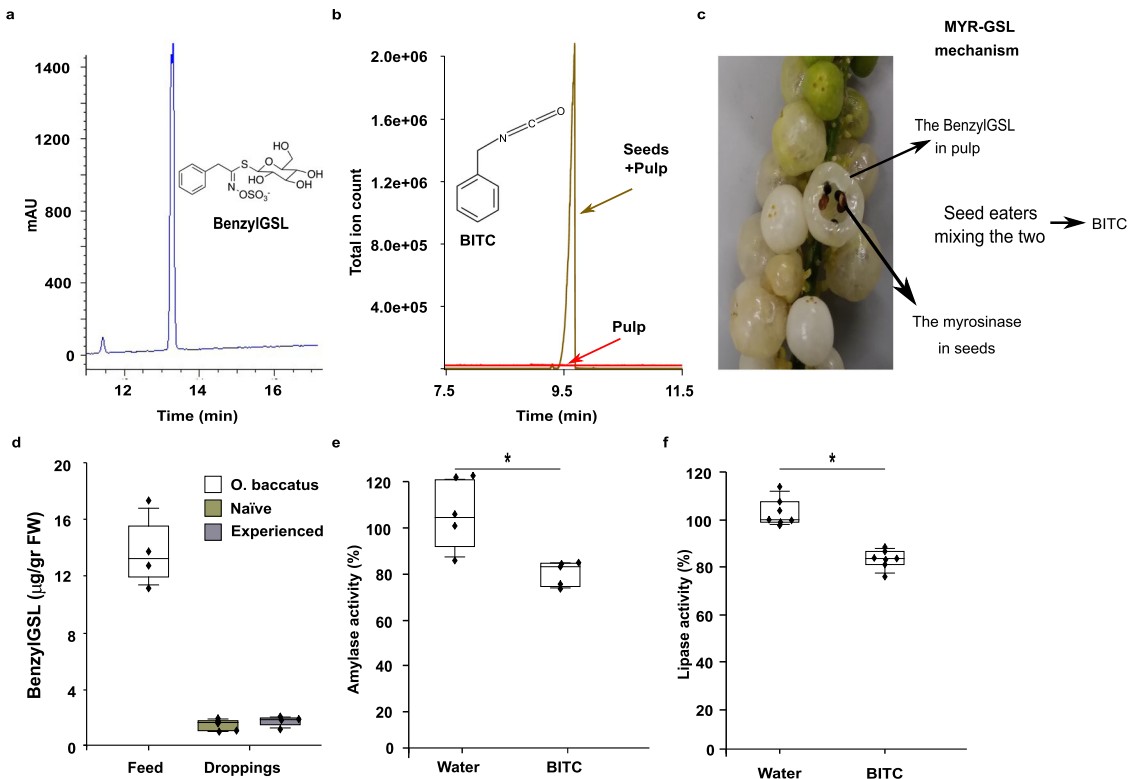

**Fig. 2 | Mixing *Ochradenus baccatus* seeds and fruit pulp activates the MYR-GSL mechanism and inhibits digestive enzymes. a** HPLC chromatogram of glucosinolates (GSLs) in *O. baccatus* pulp juice identifies BenzylGSL as the main GSL. **b** Mixing *O. baccatus* pulp juice with ground seeds leads to the production of Benzyl isothiocyanate (BITC) as the degradation product, but no BITC is found in the pulp. **c** Schematic model of the myrosinase-glucosinolates mechanism in *O. baccatus* fruits. BenzylGSL is located in the fruit pulp, whereas the myrosinase (MYR) enzyme is found only in the seeds. Mixing of these two (such as in bird digestive systems) leads to the accumulation of BITC. **d** Concentration of BenzylGSL (μg/gr FW) in fresh *O. baccatus* fruits (here Feed) and in droppings of experienced and naïve birds fed on *O. baccatus* fruits quantified by HPLC (for details on birds see Fig. 1a). Box centerlines are the median, box limits are the lower quartile (Q1) and upper quartile (Q3), and whiskers are 5–95 percentiles. One-way ANOVA, $F_{2,11} = 105.58$, $P < 0.001$, for *O. baccatus* vs. birds. Bonferroni Multiple Comparison Test, $P < 0.0001$−denoted by bars labeled with different letters; $n = 4$ independent naïve birds or 6 independent experienced birds, respectively. **e** Effect of BITC on amylase activity as compared with water; box centerlines, limits and whiskers are as in (**a**); two-tailed *t*-test for two independent samples, *$t_8 = 3.514$, $P = 0.008$, $n = 5$ for each treatment. **f** Effect of BITC on lipase activity as compared with water; box centerlines, limits and whiskers are as in (**a**); two-tailed *t*-test for two independent samples, *$t_{12} = 5.326$, $P < 0.001$, $n = 7$ for each treatment. Source data are provided as a Source Data file.

mechanism of BITC for seed protection. It should be noted that to avoid the slaughter of many birds, commercial (porcine pancreas) enzymes were used.

## Both bird diet and natural history determine the prevalence of fruit-associated *Pantoea* bacteria

The ability of birds acclimated in captivity to *O. baccatus* to digest food containing *O. baccatus* pulp and seeds (Fig. 1c) suggests an "acquired tolerance" mechanism to BenzylGSL hydrolysis products. Such tolerance can be the result of specialized digestive enzymes (as was shown for monoterpenes[28]) but is unlikely since the distance between the two capture sites is only 95 km that is continuously inhabited by *P. xanthopus* populations, making genetic mixing of the bird population highly likely. Thus, hypothesizing that this tolerance is based on the activity of gut bacteria, we examined the effect of *O. baccatus* diet on the overall composition of and specific taxa abundance in bird gut microbiota. We used next-generation 16S rRNA gene sequencing of DNA extracted from the droppings of birds feeding on *O. baccatus* fruits or banana diets (Fig. 1b, W1−3). For comparison, we also extracted and sequenced bacterial DNA from the *O. baccatus* and banana feeds. Overall, we detected 1091 amplicon sequence variants (ASVs) in the bird droppings and 233 ASVs in the fruits. We found a significant effect of capture location (naïve vs. experienced; Fig. 3a, triangle vs. circle, respectively) and diet (banana vs. *O. baccatus* fruits; Fig. 3a, yellow vs. blue, respectively) on birds microbiota composition, as demonstrated

by non-metric multidimensional scaling analysis (NMDS; Fig. 3a). Interestingly, samples from experienced birds that were switched to *O. baccatus* fruit diet clustered together with samples from experienced birds feeding on *O. baccatus* fruits (Fig. 3a) and were separate from banana-fed naïve birds, demonstrating a strong diet effect.

Looking at the genus level community composition, *Pantoea* were the most abundant bacteria in birds feeding on *O. baccatus* fruits, regardless of their geographical origin (Fig. 3b). Among *Pantoea*-annotated ASVs, ASV0003 was by far the most abundant, corresponding to >93% of total reads assigned to that genus. In sharp contrast, in birds feeding on the banana diet, the *Pantoea* genus as a whole was only 0.67% of the bacterial population (Fig. 3c), with ASV0003 representing 58% of *Pantoea* ASVs (0.4% of total bacteria). Highly consistent with this finding, the *Pantoea* were also the most abundant bacteria in the *O. baccatus* fruit microbiota (47.97% of ASVs; Fig. 3c), with AVS0003 comprising 91.4% of the total *Pantoea* ASVs.

Time course analysis of the effect of diet (*O. baccatus* fruits or banana) and its shift on the prevalence of *Pantoea* in the bird droppings supports the notion that *Pantoea* in the bird gut is closely associated with feeding on *O. baccatus* fruits (Fig. 3d). At the beginning of the experiment, *Pantoea* ASV0003 was found only in experienced birds and was completely absent from the naïve bird microbiomes (Fig. 3d). Switching diets resulted in a dramatic, rapid change in the abundances of these bacteria, increasing in birds that were switched from banana to *O. baccatus* and decreasing in birds that were switched from *O. baccatus*

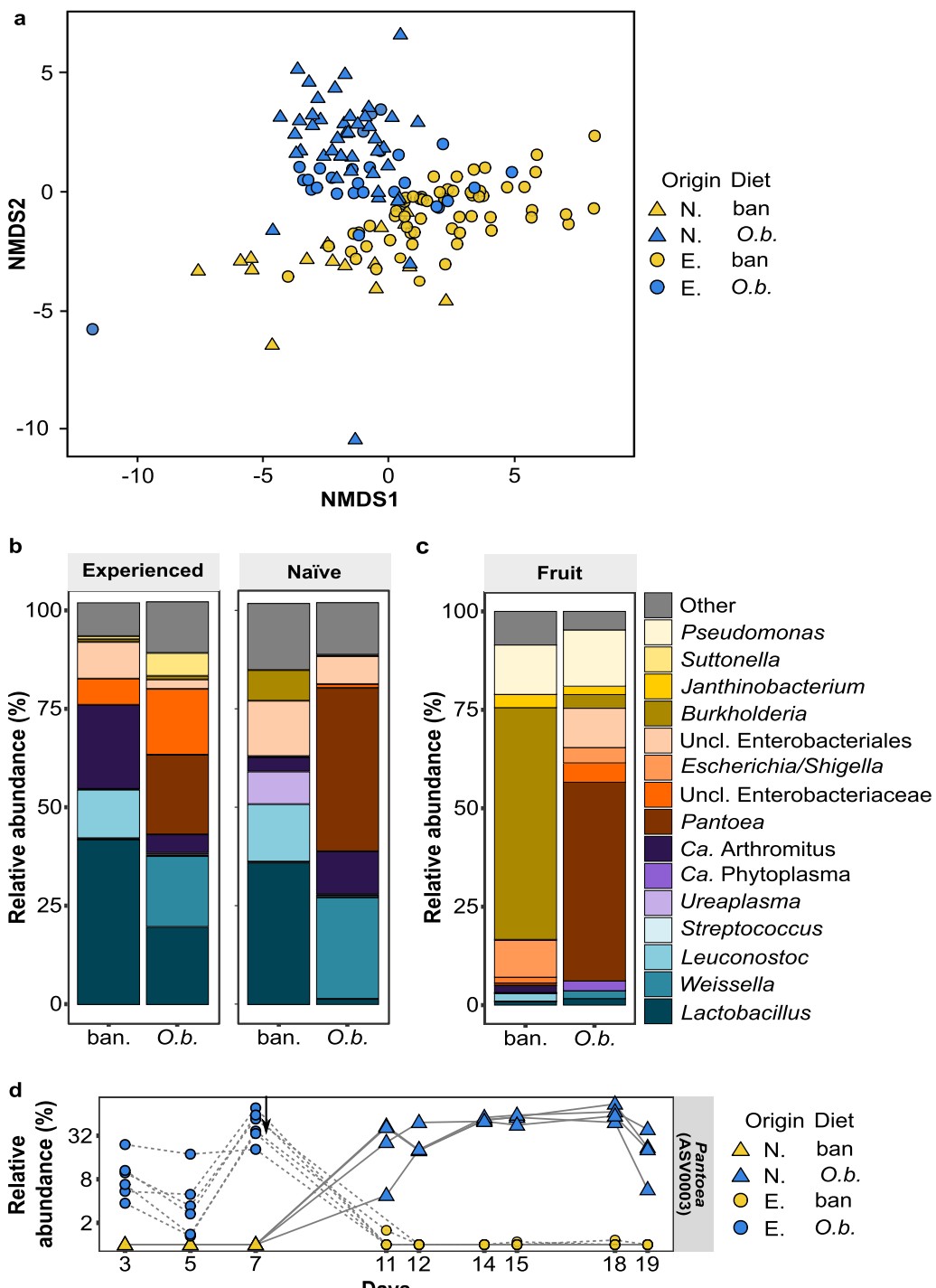

**Fig. 3 | The bird gut microbiota and relative abundance of *Pantoea* are strongly affected by the consumption of *Ochradenus baccatus* fruits. a** The difference in the microbiota composition of naïve (N.; triangles) and experienced (E.; Circles) birds feeding on banana (ban.; Yellow) or *O. baccatus* fruits (*O.b.*; Blue). Non-metric multidimensional scaling analysis (NMDS) based on the Bray–Curtis dissimilarity matrix (Stress $_{k=2}$ = 0.12) reveals that capturing location and diet have a strong effect on shaping bacterial populations; permutational analysis of variance (PER-MANOVA), $F = 6.67$, df = 1, $R^2 = 0.074$, $P < 0.001$ and $F = 7.12$, df = 1, $R^2 = 0.079$, $P < 0.001$ for capturing location and diet, respectively. **b** Mean relative abundance of bacterial genera in experienced and naïve birds fed on bananas or *O. baccatus* fruits (see Fig. 1b for details on diet regime). **c** Mean relative abundance of bacterial genera in the fruits. **d** Change in the relative abundance of *Pantoea* ASV0003 (as a percentage of total bacterial sequences) per bird during different feeding days. Each marking is one bird: naïve (N. birds are marked with a circle; experienced (E.) birds with a triangle; birds feeding on *O. baccatus* fruits in blue and birds feeding on bananas in yellow). The arrow denotes the time of the diet switch; two-tailed linear discriminant analysis (LDA) effect size = 5.21, $P < 0.0001$. Source data are provided as a Source Data file and NGS data are available under BioProject PRJNA869874.

to banana within 3 days (Fig. 3d). These results demonstrated that feeding on *O. baccatus* fruits promotes *Pantoea* in the bird gut microbiome and that these bacteria are transient rather than core members of the bird gut microbiome, originating from the *O. baccatus* fruits.

**_Pantoea_ bacteria growth is improved in the presence of BITC**
The elevated level of *Pantoea* bacteria in the *O. baccatus* fruits (Fig. 3c) and in droppings of birds feeding on these fruits (Fig. 3c, d) hints that this genus might be selected for by the presence of BITC, the

breakdown product of BenzylGSL by the MYR enzyme. To test this, we cultured bacteria from fresh droppings of experienced birds on the day of their capture (Fig. 1a: cultivatable bacteria isolation) and found that some isolates were able to grow in Luria Broth (LB) supplemented with 0.1% BITC. Of the 50 morphologically representative isolates tested, most showed reduced growth as compared to those grown on plain LB. However, a few, and especially isolate #45, showed increased growth in the presence of BITC (Fig. 4a). After DNA extraction, amplification, and sequencing of the 16S rRNA gene of isolate #45, we found it to be 99.78% identical to *Pantoea agglomerans* and 100% identical to ASV0003 (Supplementary Figs. S1 and S2, respectively; hereafter *Pa45*), the most dominant taxa in *O. baccatus* fruit and in the gut microbiome of birds fed on *O. baccatus* fruit (Fig. 3b, c). These results suggest that the *Pantoea* not only originates from the *O. baccatus* fruits but is also selected for by GSL-MYR breakdown products in the *O. baccatus* fruit.

### *Pa45* can detoxify BITC and improves digestion in birds

We examined the ability of *Pa45* (isolate #45 identified as *P. agglomerans*) to detoxify BITC. A GC-MS analysis demonstrated that the addition of Pa45 bacterial isolate to LB medium supplemented with 0.1% BITC led to a significant reduction in BITC concentration as compared to the medium without the bacteria (Fig. 4b, c). In agreement, the addition of *Pa45* significantly neutralized the inhibition effect of BITC on amylase and lipase activity (Fig. 4d, e, respectively).

To test the effect of this bacterium in vivo, we enriched the diets of birds with *Pa45* bacteria. Consistent with our in vitro finding, feeding banana-acclimated birds on banana + pulp + seeds resulted in reduced digestibility (as compared to *O. baccatus*-acclimated birds), while enrichment of the diet with *Pa45* improved digestibility, making it comparable to the *O. baccatus*-acclimated birds (Figs. 1c and 4f). Taken together, we identified a fruit-associated bacterium that neutralizes the inhibitory digestive effect of seed-pulp mixing by removing the GLSs hydrolysis products of the MYR-GSL system.

### BITC inhibits *O. baccatus* seedling development and *Pa45* can remediate the BITC effect

*Ochradenus baccatus* pulp was previously shown to have detrimental effects on the germination of *O. baccatus* seeds[15]. Here we show that supplementing irrigation water with BITC, the main hydrolysis product of the pulp GSLs (Fig. 2b), had deleterious effects on *O. baccatus* seed development (Fig. 5; +BITC) and that the addition of *Pa45* can remediate this effect (Fig. 5; +BITC+*Pa45*). In the absence of BITC, the addition of *Pa45* had no effect on seed development (Fig. 5; +*Pa45*), suggesting the bacterial effect was due to the neutralization of the BITC.

## Discussion

Previous research exploring the effect of microbiota on fruit-frugivore interaction has suggested that, at least in some cases, microorganisms

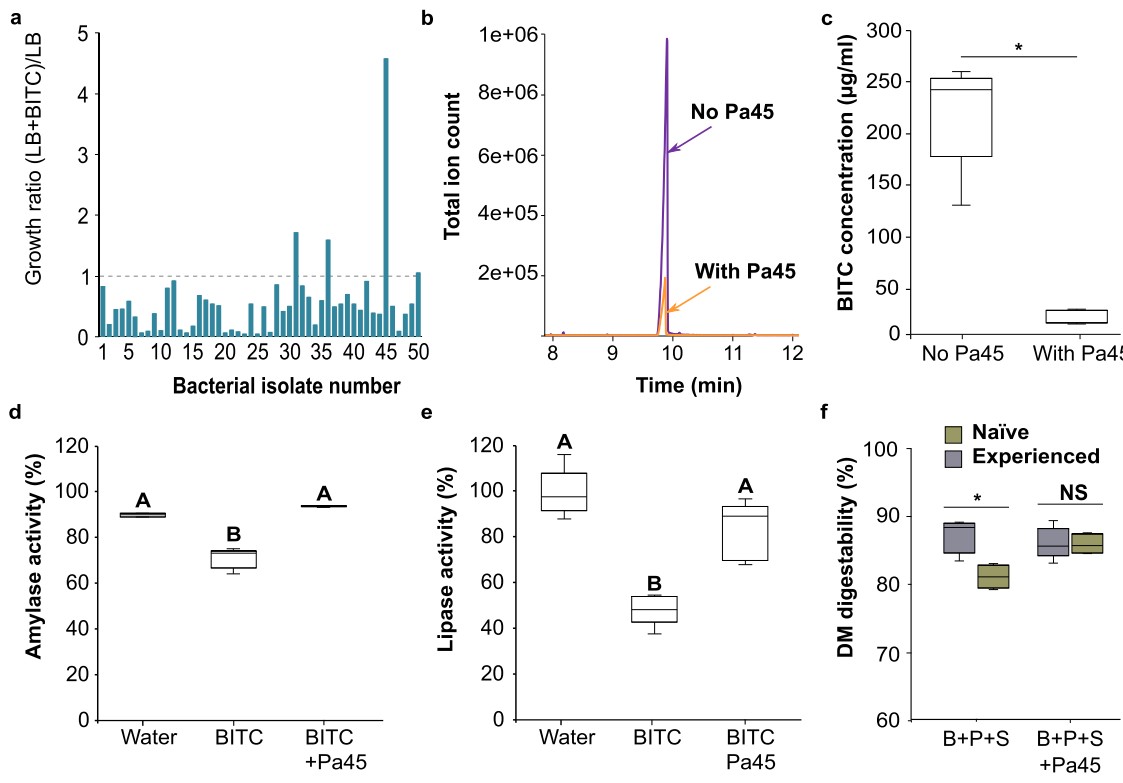

**Fig. 4 | *Pantoea* (isolate 45, 100% sequence similarity to ASV0003, see Supplementary Fig. S2) grow and remove BITC and can remediate inhibition of digestive enzymes and improve digestibility in birds feeding on a mixture of *O. baccatus* seeds and pulp. a** Screening bacterial colonies extracted from the bird droppings for the ability to grow in the presence of BITC. The results represent the growth ratio on Luria Broth (LB) medium supplemented with BITC (0.1%) relative to medium supplemented with water. The dashed line represents the potential growth on the non-supplemented medium. **b** The mixing of BITC and *Pantoea* 45 bacterial isolate (*Pa45*) results in the removal of BITC as can be seen in the representative GC-MS chromatogram. **c** Quantification of BITC removal by *Pa45*; Box centerlines are the median, box limits are the lower quartile (Q1) and upper quartile (Q3), and

whiskers are 5–95 percentiles; two-tailed *t*-test for two independent samples, $t_8 = 9.03$, $P < 0.0001$, $n = 5$ for each treatment. **d** Effect of BITC, BITC+*Pa45* and water on amylase activity; box centerlines, limits and whiskers are as in (**c**). One-way ANOVA, $F_{2,12} = 110.14$, $P < 0.0001$, $n = 5$ for each treatment. **e** Effect of BITC, BITC +*Pa45*, and water on lipase activity; box centerlines, limits and whiskers are as in (**c**). One-way ANOVA, $F_{2,12} = 32.84$, $P < 0.0001$, $n = 5$ for each treatment. **f** Dry matter digestibility (%) of experienced birds and naïve birds fed on a mixture of ground *O. baccatus* seeds and pulp juice before or after enrichment with *Pa45*; box centerlines, limits and whiskers are as in (**c**); two-tailed *t*-test for two independent samples, $t_8 = 4.306$, $P = 0.003$, $n = 4$ naïve birds and 6 for experienced birds. NS not significant. Source data are provided as a Source Data file.

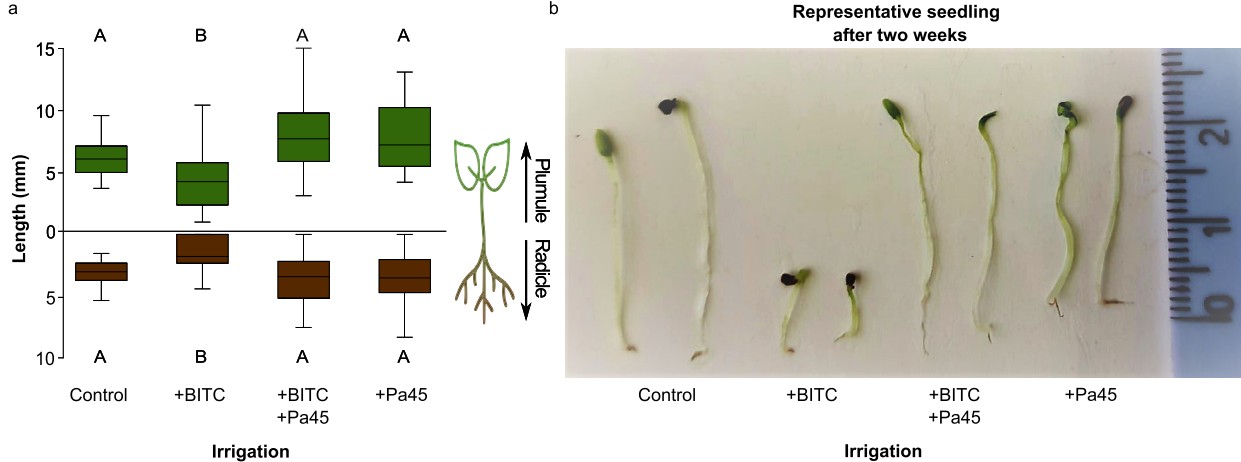

**Fig. 5 | Benzyl isothiocyanate inhibits *Ochradenus baccatus* seedling growth and *Pa45* neutralizes inhibition. a** Plumule (green) and radicle (brown) growth (length, mm) after irrigation with water relative to water supplemented with BITC (0.1%) in the presence or absence of *Pantoea* 45 bacterial isolate (*Pa45*); box centerlines are the median, box limits are the lower quartile (Q1) and upper quartile (Q3), and whiskers are 5–95 percentiles. Plumule: one-way ANOVA, $F_{3,142} = 13.21$, $P < 0.001$, $n = 30$ to 54 for each treatment. Radicle: one-way ANOVA, $F_{3,142} = 7.21$, $P < 0.001$, $n = 30$–54 for each treatment. **b** Representative seedling after irrigation with water vs. water supplemented with BITC (0.1%), in the presence or absence of *Pa45* after 2 weeks of growth. Source data are provided as a Source Data file.

can enhance the attractiveness of fruits to vertebrate herbivores[29]. Here we explored this interaction for *Ochradenus baccatus*, a desert shrub that produces fleshy fruits, having overcome seed digestion by frugivores. Besides sugar content, the fruit pulp contains glucosinolates and the seeds contain the myrosinase enzyme. When the seeds and pulp are mixed; for example, in an animal's digestive tract, toxic hydrolysis products are released by the MYR-GSL mechanism. Previously, this formation of hydrolysis products was shown to encourage rodents to shift from seed predation to seed dispersal[23]. *Pycnonotus xanthopygos* birds (bulbuls) are considered important seed dispersers of *O. baccatus*, consuming the whole fruit and secreting the seeds[26]. Nevertheless, our previous study demonstrated a complex interaction between the birds and the plant where passage through the bird's digestive system dramatically reduced the number of viable seeds (and thus reduced plant fitness). However, the digestive passage also increased the germination of the surviving seeds[15] but hampered the bird's digestive ability by inhibiting digestive enzyme activity, demonstrating the existence of conflicting interests and an ecological arm race[15].

In the current study, we decipher the mechanism behind these effects, showing that activation of the "*O.* baccatus-oil bomb" (i.e., the GSL-MYR system) produces BITC that inhibits bird digestion and digestion enzymes and *O. baccatus* seed growth. We also show that a bacterium, *Pantoea agglomerans*[30], found to be a common resident in *O. baccatus* fruits, and acquired from fruit intake by *P. xanthopygos*, could reduce concentrations of BITC. Furthermore, *P. xanthopygos* can remediate the physiological consequences of BITC exposure and fully restore bird digestion and seedling growth. These results fit well with the previously suggested role of the gut microbiome in detoxification of "pollutants" in food[31] for insects[32], rodents[33] and sunbirds[34], but here we demonstrate that a specific bacterial species can mediate a multitrophic interaction positively affecting both bird digestive physiology and plant demographics (Fig. 6). We also show that *P. agglomerans* is a transient member of the *P. xanthopygos* gut microbiome, established and stabilized in this environment by the consumption of *O. baccatus* by the birds. Switching the birds' diet from *O. baccatus* fruits to bananas resulted in a dramatic and rapid decrease of the *Pantoea* population in the bird gut, and vice versa when switching from bananas to the *O. baccatus* diet (similar to refs.[35,36]). As an isolate from the

bird droppings, the same *P. agglomerans* showed not only resistance but also improved growth in the presence of GSL hydrolysis products, when most other isolates from the same droppings showed reduced growth as such breakdown products are known to harm many bacteria[37] (but see refs.[38,39] for previous studies on resistant bacteria).

In summary, the data presented here offer an explanation for the effect of glucosinolates breakdown products on the digestive ability of vertebrates and the detoxification of these products by a specific bacterial species, *P. agglomerans*, acquired from consumed fruits, and selected for in the presence of glucosinolates hydrolysis products. The bacteria can benefit from these interactions by gaining a unique, SM-protected habitat in both the bird intestine and in the *O. baccatus* fruits, extending the reach of the latter. Given that *Pantoea* bacteria are common in many environments, and that glucosinolates are common secondary metabolites, these bacteria may be involved in many other plant–animal interactions (e.g., ref.[40]). Finally, our results highlight the complexity of conflicting interests between different tripartite kingdom members: a plant, a fruit consumer, and a specific bacterial species.

## Methods

This study received and was conducted according to the University of Haifa ethics committee approval number 750/21.

### Study species

*Ochraceous baccatus* (Delile 1813; Resedaceae) is a perennial wild shrub with wide Saharo-Sindian distribution and is found in desert habitats from Pakistan to central Israel[41]. The data for *O. baccatus* range in Israel (Fig. 1a) were taken from Danin and Fragman-Sapir (2016) Flora of Israel Online https://flora.org.il/en/plants/OCHBAC/#moreinfo). All *O. baccatus* fruit used in the experiments were harvested near the northern coast of the Dead Sea, Israel (31°48′N, 35°27′E) from various shrubs. Fresh fruits were collected once every 3 days, mixed and stored at 4 °C until use. Where required, *O. baccatus* seeds and pulp were manually separated. The pulp was crushed and squeezed through filter paper for juice. Where designated, seeds were ground in short pulses in a coffee grinder, and the seed powder was suspended in water at 20% wt/vol (e.g., 2 g ground seeds in a total water volume of 10 mL). The pulp juice and seed extract were centrifuged (8000 × g, 5 min) to

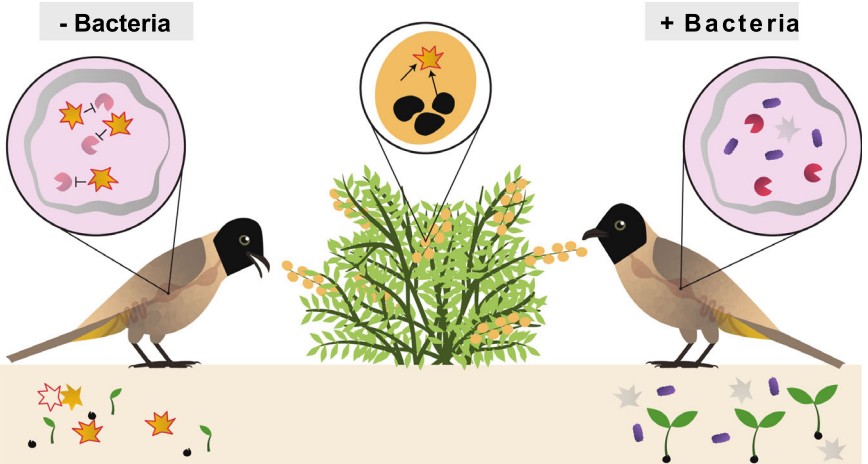

**Fig. 6 | The effect of the "*Ochradenus baccatus*-oil-bomb" on the bird and on seedling growth in the presence of *Pantoea* 45 bacterial isolate.** A schematic model describing the multitrophic interaction mediated by the bacteria, positively affecting both plants and birds. After mixing of *O. baccatus* pulp and seeds, the "*O. baccatus*-oil-bomb" is activated and BenzylGSL is created (star) in the bird's digestive system, leading to the inhibition of digestive enzyme activity (purple PacMans), and thus reduced digestion. Furthermore, the products of the "*O. baccatus*-oil-bomb" also cause a decrease in seedling growth. *Pantoea* bacteria found in *O. baccatus* fruits is acquired by the birds, as observed in their gut microbiota, and neutralizes the "*O. baccatus*-oil-bomb", thus facilitating the bird's digestion and restoring seedling growth.

remove large solids and sterilized by filtration through 0.2 μm syringe filters.

*Musa acuminata* Colla 1798 (banana) is an evergreen perennial plant belonging to the family Musaceae. Its fruits are often used as a main component in artificial diets of frugivorous birds[42]. Banana pulp contains high amounts of soluble carbohydrates (70% of dry weight) but low levels of protein (1.2%) and lipids (0.2%)[43]. In our experiments, we used the Grand Nain variety, the most common cultivar grown in Israel.

*Pycnonotus xanthopygos* (Hemprich and Ehrenberg 1833; white-spectacled bulbuls) is a common Middle East resident[44]. *P. xanthopygos* were captured from two habitats: one overlapping with the *O. baccatus* range (31°33' N, 35°24' E), and thus assumed to be unexperienced in foraging and consuming *O. baccatus* fruit[24,26] (i.e., experienced birds; *n* = 6) and the second outside the *O. baccatus* distribution area (32°27' N, 35°33' E), and thus assumed to be naïve to *O. baccatus* fruit (i.e., naïve birds; *n* = 4). Birds were captured using mist nets (Monofilament 9S; mesh size 15 × 15 mm, net size 2.5 × 12 m) on May 2018 at the Jordan Valley, Israel (bird ringing permit #297, Israel Nature & Park Authority). The birds' ages were 6 months to 1 year. Sex determination in *Pycnonotus xanthopygos* is very complex without surgery, which we did not do. All birds were released unharmed at the end of the experiment.

### Bird capture, housing and diet
After capture, each bird was kept in a separate cage (100 × 80 × 40 cm) in an outdoor enclosure at the Oranim College Campus, Kiryat Tivon, Israel. Upon capture, feeding commenced. In the first week, naïve birds were fed on banana fruit, while experienced birds were fed on *O. baccatus* fruit (Fig. 1b, W1). At the beginning of the second week, and for 2 weeks (W2–3), naïve birds were fed *O. baccatus* fruit while experienced birds were fed on banana fruit. During these 3 weeks (Fig. 1b, W1–3), droppings and fruit samples were collected daily and placed in sterile tubes, and two volumes of ethanol were added. Tubes were frozen at −20 °C. Next, at the beginning of week 4 (Fig. 1b, W4), diets were switched back, and until the end of the experiment, naïve birds were fed on bananas, while experienced birds were fed on *O. baccatus* fruit. At specified intervals (Fig. 1b), one meal of the birds' diet was supplemented with either water, *O. baccatus* fruit pulp juice (containing glucosinolates), *O. baccatus* ground seed extract (containing the myrosinase enzyme) or both pulp juice and seed extract (activating the *O. baccatus* GSL-Myr system and resulting in GLS breakdown products).

### Quantification of digestion ability
Digestibility was quantified as in ref. [16]. Briefly, on the morning of each test (07:00 am), water and food containers were removed from all cages. At 11:00 am, baking paper sheets were placed at the bottom of each cage to collect droppings and a container holding 50 g (fresh weight) of feed mixture (see below) was presented to the birds. The birds were allowed to feed ad libitum for 30 min, after which the dishes were removed. The food remaining in the containers was weighed and the dry weight of the food consumed (gram dry weight) was calculated according to Eq. 1. In all cases, dry weight was determined by drying known amounts of feed at 60 °C to stable weight and calculating the dry mass percentage. In addition, fresh food weight loss due to evaporation was quantified by placing five containers (similar to the containers presented to the birds, containing 50 g of each food) out of the cages during the feeding period and weighing the food after the feeding period (evaporation loss was less than 6% in all cases). Change in food weight was corrected for food weight loss due to evaporation. The 50-g feed was supplemented with 2 mL water (control); 1 mL *O. baccatus* pulp juice + 1 mL water (pulp, P); 1 mL *O. baccatus* ground seed extract + 1 mL water (seeds, S); or 1 mL *O. baccatus* pulp juice + 1 mL of *O. baccatus* ground seed extract (pulp + seeds, P+S).

The maximal gut passage time of *O. baccatus* fruits in *P. xanthopygos* is 91.5 ± 7.3 min[26]. Therefore, 2 h after the end of feeding (i.e., 13:30 pm), the linings on the cage floors were collected and the droppings scraped into pre-weighed pre-labeled glass Petri dishes (the glass used as "plastic" tended to lose weight do to evaporation of plasticizers). The droppings were dried to a constant weight at 60 °C and food digestibility was calculated according to Eq. 2. After each experiment, the naïve birds were returned to feed on banana and the experienced birds were returned to feed on *O. baccatus* fruit ad libitum. All birds were seen to be in good health during the experiments and willingly consumed all feeds.

$$\text{Food consumption}_{\substack{(\text{g dry wt})}} = \left[ \text{food supplied}_{\substack{\text{g fresh wt}}} - \frac{\text{remaining food(g fresh wt)}}{1 - \text{percentage food loss due to evaporation}} \right] \times \% \, \text{dry wt}$$

(1)

$$Digestability(\%) = \frac{(food\ consumption\ (g\ dry\ wt) - droppings(g\ dry\ wt))}{food\ consumption(g\ dry\ wt)} \times 100 \quad (2)$$

### Identification of *O. baccatus* fruit glucosinolates

Pulp and seeds of *O. baccatus* fruit stored at 4 °C were separated manually. Three hundred mg of pulp were transferred to an Eppendorf tube containing 1200 μL methanol and 100 μM sinigrin (as internal standard; no sinigrin was found in *O. baccatus*[45]). The GSLs were extracted on a DEAE column, converted to desulfoGSLs, and identified by HPLC by comparing the retention time and the absorbance spectrum (all according to ref. [45]).

### Quantification of BenzylGSL in fresh *O. baccatus* fruits and the droppings of birds fed on *O. baccatus* fruit

Three hundred mg of fresh *O. baccatus* pulp were placed in an Eppendorf tube containing 1200 μL methanol and 100 μM sinigrin (as internal standard). Simultaneously, 300 μL of droppings from birds fed on *O. baccatus* fruit (experienced birds in the first day capture and naïve birds after 1 week when the diets were switched) were also transferred to an Eppendorf tube containing 1200 μL methanol and 100 μM sinigrin (as internal standard). The GSLs were extracted, converted to desulfoGSLs, identified, and quantified by HPLC according to ref. [45].

### Identification of *O. baccatus* glucosinolates-myrosinase hydrolysis products

To identify GSL hydrolysis products, we used gas chromatography combined with mass spectrometry (GC-MS). Forty mL of ground *O. baccatus* fruit (pulp and seeds) were placed in a 100 mL Erlenmeyer flask and overlayered with 10 mL of n-Hexane for 2 h. Then, the liquids were transferred to 50 mL tubes, mixed by vortex for 1 min, and centrifuged at 4000 × g for 5 min to remove solids. Next, 5 mL of the upper organic phase was transferred to a glass tube and concentrated to 100 μL under a nitrogen stream. The concentrated solution was transferred to a glass GC vial and 2 μL were injected into an Agilent 7890A gas chromatograph (Agilent Technologies), equipped with a Rxi-5Sil MS capillary column (30 m length; 0.25 mm ID; 0.25 μm coating; Restek) and coupled to a single-quad mass spectrometer detector (Agilent 5975C). The initial oven temperature was 70 °C. It was raised at a rate of 20 °C/min to 250 °C and then at 10 °C/min from 250 °C to 320 °C and held for 10 min. The injector temperature was set at 250 °C and the carrier gas (helium) flow was 1 mL/min. Splitless injection was used with a purge time of 1 min. The transfer line was set at 320 °C. Mass spectrometer conditions were as follows: ion source temperature at 230 °C, quadrupole temperature of 150 °C, and ionization voltage of 70 eV. Mass analysis was performed with an electron impact source under scan mode from 40 to 550 m/z. Identification of GSL hydrolysis products was based on retention time and comparison of fragmentation patterns with the National Institute of Standards and Technology (NIST) library eleven and validated against a commercial standard (Sigma Cat# 622-78-6).

### Effect of BITC on digestive enzymes activity

To test if BITC could inhibit digestive enzymes activity, commercial digestive enzymes amylase (Sigma Cat# A3176) and lipase (Sigma Cat# L0382) were tested after being combined with aqueous commercial BITC (0.2%; Sigma Cat# 622-78-6). Deionized water was used as a control. Amylase and lipase assays were performed according to ref. [15].

### DNA extraction from bird droppings and plant material, and 16S rRNA gene amplification and sequencing

To study the effect of the birds' origin and diet on bacterial community composition, we analyzed the microbiota from droppings of the birds from two habitats, fed as described above (also see Fig. 1b). Bird

droppings (-500 μL) were collected daily into a sterile tube, and two volumes of absolute ethanol were added. In addition, bananas and *O. baccatus* placed outside the cages were collected daily at the time of feeding to analyze fruit microbiota. All samples were stored at −20 °C until DNA extraction. Sample tubes were centrifuged for 10 min at 15,000 × g and the ethanol was removed by pipetting, followed by evaporation for 20 min in a 100 °C dry block. DNA was extracted from the solids using PureLink™ Microbiome DNA Purification Kits (Thermo Fisher Scientific, UK) following the manufacturer's instructions. DNA was also extracted from three empty tubes that went through the same treatment as negative controls. The extracted DNA was kept at −20 °C until further analyses. For PCR amplification, 10–100 ng DNA was amplified for the V4 region of 16S rRNA gene using the primer pair CS1_515F (ACACTGACGACATGGTTCTACAGTGCCAGCMGC CGCGGT AA) and C1_806R (TACGGTAGCAGAGACTTGGTCTGGACTACHVG GGTWTCTAAT) (Integrated DNA Technologies, USA) as previously published[46]. Amplification was performed in a volume of 25 μL with the 2× Bio-Ready-Mix (Bio-Lab, Israel). Primer concentrations were 0.5 ng/μL. PCR conditions were: 95 °C for 5 min, followed by 28 cycles of 30 s at 95 °C; 45 s at 55 °C and 30 s at 68 °C, and a final elongation step of 7 min at 68 °C. The amplification products were verified by agarose gel electrophoresis and then stored at −20 °C. No amplification (contamination) was observed in the blank samples.

Library preparation and sequencing were performed at the Genome Research Core, Research Resources Center, University of Illinois, Chicago, USA. Briefly, a second PCR amplification was performed for each sample to introduce a unique 10-base barcode obtained from the Access Array Barcode Library for Illumina (Fluidigm, CA, USA; Item# 100-4876). Samples were pooled in equal volume using and purified using an AMPure XP cleanup protocol (0.6×, vol/vol; Agencourt, Beckmann-Coulter). The pooled libraries, with a 15% PhiX spike-in, were loaded onto an Illumina MiniSeq mid-output flow cell (2 × 150 paired-end reads) to generate sequence data. Raw reads were recovered as FASTQ files. Raw sequence data were deposited in the NCBI SRA database under BioProject accession PRJNA869874.

### Sequence data processing and analysis

Sequence analysis—the DADA2 pipeline (version 1.18.0)[47] using R was used for sequence data analyses. Sequences were filtered and trimmed for quality using the "filterAndTrim" command with the parameters maxN = 0, maxEE = 2, trimLeft = 20 bp and truncLen = 153 bp. A sequence error estimation model was carried out using the "learnErrors" option using default parameters. Following these steps, the DADA2 algorithm was implemented for error correction, and a count table containing the ASVs and counts per sample was produced. Sequences were merged using the "mergePairs" command with minimum overlap set at 8 bp. Suspected chimeras were detected and removed using the command "removeBimeraDenovo". Count tables with ASV sequences and the number of reads per sample were extracted. Taxonomic classification of ASVs was assigned using the SILVA small subunit *rRNA* database (version 138; https://zenodo.org/record/3731176#.Y9ZyjnZBxaQ), using the command "assignTaxonomy" algorithm with a minimum similarity threshold of 80% and the Phyloseq package (version 1.34.0) used to create the ASVs table. Sequences that were assigned as non-bacterial or unclassified (e.g., Eukaryotes, Chloroplast, and Mitochondria) were removed. Then, we excluded bird fecal samples with total read counts below 3500. Due to the high rate of chloroplast-originating sequences in plant samples, we analyzed all plant samples with bacterial read counts above 1000.

Count data were normalized by the cumulative sum square method (CSS)[48]. In order to examine the effect of bird origin (naïve vs. experienced) and diet (banana or *O.b.*) on the microbial community composition, the permutational analysis of variance (PERMANOVA) test was applied. PERMANOVA was performed using the command "adonis2" in R package "vegan" (version 2.6-2)[49]. In addition,

non-metric multidimensional scaling (NMDS) analysis was done in "vegan" based on Bray–Curtis dissimilarities calculated from CSS normalized counts. NMDS was done using the command "metaMDS" in "vegan" with parameters $k = 2$, try = 1000. Relative abundances for ASVs and genera were calculated based on CSS normalized counts. Linear discriminant effect size (LEfSe) analysis was chosen to calculate differential abundance and identify biomarkers for bird diet. This method is effective in determining which features, in this case ASVs, are most likely to explain observed differences among factor levels[50]. LEfSe was performed using the online Galaxy module (http://huttenhower.sph.harvard.edu/galaxy).

### Isolation of cultivatable bacteria from experienced bird droppings and their identification

On the first day of captivity, 500 μL of droppings were collected from each experienced bird and placed into a sterile tube as described above but without the addition of ethanol. The droppings were immediately plated on Luria Broth (LB) agar and incubated at 28 °C for 24–48 h. Representative colonies were selected by morphology and subcultured five times to achieve purity for each isolate. Cultures of each isolate, grown overnight in LB were used for DNA extraction and PCR, as well as biochemical testing. Genomic DNA was purified from cultures using the DNA Fungal/Bacterial Microprep Kit (Zymo Research, USA) according to the manufacturer's instructions. The near full-length 16S rRNA gene was amplified by PCR using primers 11F (5′-GGATCCAGACTTTGATYMTGGCTCAG-3′) and 1512R (5′-GTGAA GCTTACGG(C/T)TAGCTTGTTACGACTT-3′), modified from ref. [51]. The thermal cycling conditions were 94 °C for 4 min; 30 cycles at 94 °C for 30 s, at 54 °C for 40 s, and at 72 °C for 70 s; and then a final step at 72 °C for 20 min. The PCR products were sequenced by MCLAB (San Francisco, CA) and the sequences were compared to those available in the NCBI nucleotide database, using the standard nucleotide–nucleotide BLAST tool (BLASTN; http://www.ncbi.nlm.nih.gov).

### The effect of BITC on isolate growth

Tubes with 3 mL of liquid LB medium supplemented with BITC (0.1% v/v) or water (as a control) were inoculated with the different bacterial isolates (see above) to O.D.$_{600nm}$ of 0.05, measured in a 1 cm cuvette using a spectrophotometer (UV-1650PC, Shimadzu). The cultures were incubated at 28 °C for 24 h. Then, the O.D.$_{600nm}$ of each culture was measured, and the ratio between growth LB medium supplemented with BITC (0.1%) relative to water was calculated.

### The effect of isolate *Pa45* on BITC amount

A 100-mL Erlenmeyer flask with 40 mL of LB was supplemented with BITC (0.1% v/v) and inoculated with *Pa45* (added to a final O.D.$_{600nm}$ of 0.05). Ten mL of n-Hexane (Merck, 110-54-3) was overlayered and the Erlenmeyer was sealed with Parafilm. A sterile medium was used as a negative control. Following 24 h of incubation at 28 °C with shaking at 200 rpm, the liquids were transferred to 50 mL tubes, mixed by vortex for 1 min, and centrifuged at 4000 × g for 5 min for phase separation. Next, 5 mL of the upper organic phase was transferred to a glass tube and concentrated to 100 μL under nitrogen flow. The concentrated solution was analyzed with GC-MS as above and BITC quantification against a calibration curve was performed on BITC dissolved in n-Hexane (1, 3 and 5 μg/mL).

### The effects of BITC on *O. baccatus* seedling growth

To assess the effect of BITC on *O. baccatus* seedling development, *O. baccatus* seeds were separated manually immediately after the collection of fruits, and they were germinated as in ref. [15]. Soaking liquids were water, water inoculated with *Pa45*, water supplemented with BITC (0.1% v/v), and water supplemented with BITC (0.1%) and inoculated with *Pa45*. The seedlings were left for 2 weeks. Seedling plumule and radicle were measured with a caliber.

### Statistical analysis

Statistical analyses were performed using SPSS statistical software version 21.0 (IBM, SPSS Inc., Chicago, IL, USA). Normality was examined using the Shapiro–Wilk test, and data were considered statistically significant for a $p$ value <0.05. Two-tailed $t$-tests for two independent samples were applied to the effect of food provided for digestion efficiency and for the amount of BITC in the presence or absence of *Pa45*. A one-way ANOVA followed by Bonferroni multiple comparisons test was done on the quantity of the GSLs in *O. baccatus* fruits before or after passage through the bird digestive tract and on the raw enzymatic activity for each enzyme.

## Data availability

The raw amplicon sequence data generated in this study have been deposited in the NCBI SRA database under accession code PRJNA869874. Raw results of feeding experiments, digestibility, enzyme activity, BenzylGSL and BITC quantification, abundance of bacterial genera, number of OTUs, and seedlings growth are given in the "Supplementary Dataset" file. Source Data are provided with this paper.

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

## Acknowledgements

We thank the Israeli Science Foundation (ISF grant No. 296/16 to I.I.) and the Middle East Regional Cooperation Program (project TA-MOU-08-M28-013 to Y.G. and I.I.) for their generous funding. We are grateful to R. Haran, who helped capture the birds, and to N. Dainov of the Oranim College animal house staff for her help with animal maintenance.

## Author contributions

Conceptualization: B.T., I.I., and Y.G. Methodology: B.T., I.I., and Y.G. Investigation: B.T. Formal analysis: B.T., M.L., and I.I. Writing—original draft: B.T., M.L., N.S., I.I., and Y.G. Funding acquisition: I.I. and Y.G. Supervision: I.I. and Y.G.

## Competing interests

The authors declare no competing interests.
