## [Peer Review File · Nature Communications]

In their review of the first version of this manuscript, reviewer #1 added some of their comments to the manuscript file. These comments have been addressed by the authors within the Peer Review File.

REVIEWER COMMENTS

Reviewer #1 (Remarks to the Author):

I am extremely excited to support the publication of a paper providing a mechanistic understanding of plant-animal-microbial interactions. The experimental design is solid with only a few minor recommendations. These represent the type of controlled studies that are grounded in an ecological system that are greatly needed to better understand the role of microbes in organisms and how they influence species interactions. Well done!

I have minor recommendations. The majority of my edits and comments are to improve the clarity of writing.

In addition to some grammatical edits (track changes in word document), my minor concerns are:

1. clarification of terms that vary in meaning between text, headings and figures and sometimes are not as descriptive and clear as they could be – suggestions as comments.
2. Lack of hypotheses and rationale in it's own section. Instead hypotheses are sprinkled around the results. I suggest stating hypotheses that you tested and include rationale for those hypotheses before the results section.

If the journal requests this format, keep the last paragraph of intro as is - but this seems like a conclusion or result and not a set up to transition into the results

3. Unless this is a requirement of the journal, I suggest some revision in organization described as comments.
4. There seem to be redundant details of methods in results, methods and figures. I suggest remove unnecessary methods from results unless the journal formatting requires this redundancy.
5. Be more precise of what the negative consequences are with eating fruits and specific about whether negative consequences are from pulp, seed, or digestion of both.
6. You did not measure fitness. And I question what it means to have "success". Instead use plant demographics and animal or bird physiology. and keep order of animals first then birds as that is order you performed experiments and presented results.
7. If the data exist, I think it could really inform results and interpretation to include differences in intake (from food intake info and known conc of BenzylGSL in food) and exposure of BITC from (from % of intake of BenzylGSL detected in droppings) and better explain Fig 1c by measuring amount of BITC in droppings. Since you did analysis of droppings, you could detect and quantify parent compound and likely also BITC in those droppings. If BITC is equal in both overlap and non-overlap droppings, then it is resistance of digestive enzymes of host, but if differ, then further supports recovery data with Pa45 in Fig 4. Fig 4 results are super compelling so not critical to include the "exposure" data, but it would help strengthen role for microbes rather than host in explaining negative affects of BITC in digestion.

I did not have any comments on the supplemental documents.

I look forward to reading this paper once published.

Sincerely,
Jennifer Sorensen Forbey
Professor Biology
Boise State University

Reviewer #2 (Remarks to the Author):

This excellent and original manuscript falls within the scope of Nature Communication and presents well-designed experiments and analyses. This manuscript focuses on presenting one of the first multi-kingdom dispersal interactions. This manuscript can attract great interest in the field of dispersal ecology. Overall, the manuscript fits well with the journal's standards. I have only a few comments for the authors. Please see them below.

Overall manuscript: Please check the whole manuscript for spelling and for odd sentences.

Line 16: Do you mean *Ochradenus baccatus*?

Line 31-33: Please rephrase this odd sentence

Line 41: „plant fitness and resulting“ the "and" is not necessary here.

Line 49: do you mean food intake?

Line 57: Please rephrase this sentence

Line 64: I would rather use the word destroy instead of "degrade"

line 81: „overlapping“ do you mean overlapping?

Line 82: missing „i“ from distribution

Line 87: „overlapping“ here again...

Line 88: correct the word influence

Line 92: missing c from acclimated

Methods:

How long the fruits were stored?

Reviewer #3 (Remarks to the Author):

In this comprehensive study, authors unravel a set of concatenated interactions among a plant, a bird and bacteria and their effects on the participant surrogate of fitness. In such a tripartite cross-kingdom interaction, using a series of well-designed experiments and measurements, authors reveal that the interaction between the plant and its avian seed disperser is critically mediated by *Pantoea* bacteria that reside in both the plant and the bird's intestine. I certainly believe that this manuscript is novel by significantly contributing to open new avenue of research on how microbes mediate plant-animal interactions; however, I miss in the text a mention to a few previous studies that are relevant in such research context (see below).

Specific comments:

L13. ...in their droppings and regurgitates.

L135. Interestingly, ...

L222-224, L244-245. Authors argue that they demonstrate for the first time that a specific bacterial species can mediate a multitrophic interaction positively affecting both the plant and the birds. In a previous study, Peris and collaborators (2017) found that vertebrate frugivores, fleshy-fruited plants, and microbes may form a tripartite cross-kingdom interaction in which each part interact positively with the other two. Specifically, they propose that orange (*Citrus sinensis*) seeds and *Penicillium digitatum* spores are both dispersed by mammals, and that fungal infection of orange fruits facilitates their encounter and exploitation by frugivores (mammals and birds). The seminal work by Janzen (1977) is also relevant in the context of plant-seed disperser interactions mediated by microbes.

Peris JE, A Rodriguez, L Peña, and JM Fedriani. 2017. Fungal infestation boosts fruit aroma and fruit removal by mammals and birds. *Scientific Reports*, Doi:10.1038/s41598-017-05643-z.

Janzen, D. H. (1977). Why fruits rot, seeds mold, and meat spoils. *The American Naturalist*, 111(980), 691-713.

REPOSENSE TO REVIEWERS' COMMENTS

Reviewer #1 (Remarks to the Author):

I am extremely excited to support the publication of a paper providing a mechanistic understanding of plant-animal-microbial interactions. The experimental design is solid with only a few minor recommendations. These represent the type of controlled studies that are grounded in an ecological system that are greatly needed to better understand the role of microbes in organisms and how they influence species interactions. Well done!

Thank you very much for your encouraging comments!

I have minor recommendations. The majority of my edits and comments are to improve the clarity of writing.

In addition to some grammatical edits (track changes in word document), my minor concerns are:

1. clarification of terms that vary in meaning between text, headings and figures and sometimes are not as descriptive and clear as they could be – suggestions as comments. - Corrected
2. Lack of hypotheses and rationale in it's own section. Instead hypotheses are sprinkled around the results. I suggest stating hypotheses that you tested and include rationale for those hypotheses before the results section. If the journal requests this format, keep the last paragraph of intro as is - but this seems like a conclusion or result and not a set up to transition into the results – Ignored par the editor advice

Thank you for this comment, we will follow the editors' instructions and maintain the current format of the introduction.

3. Unless this is a requirement of the journal, I suggest some revision in organization described as comments.

4. There seem to be redundant details of methods in results, methods and figures. I suggest remove unnecessary methods from results unless the journal formatting requires this redundancy. - Corrected

5. Be more precise of what the negative consequences are with eating fruits and specific about whether negative consequences are from pulp, seed, or digestion of both. The lesser digestion is the negative consequences emerging from mixing the pulp and the seeds. We have clarified that in the text as suggested by the reviewer.

6. You did not measure fitness. And I question what it means to have "success". Instead use

plant demographics and animal or bird physiology. and keep order of animals first then birds as that is order you performed experiments and presented results. - We agree with this comment and have corrected the term in the text

7. If the data exist, I think it could really inform results and interpretation to include differences in intake (from food intake info and known conc of BenzylGSL in food) and exposure of BITC from (from % of intake of BenzylGSL detected in droppings) and better explain Fig 1c by measuring amount of BITC in droppings. Since you did analysis of droppings, you could detect and quantify parent compound and likely also BITC in those droppings. If BITC is equal in both overlap and non-overlap droppings, then it is resistance of digestive enzymes of host, but if differ, then further supports recovery data with Pa45 in Fig 4. Fig 4 results are super compelling so not critical to include the “exposure” data, but it would help strengthen role for microbes rather than host in explaining negative effects of BITC in digestion. - We have attempted to quantify BITC in the droppings but failed to detect it. We did not test it here but previously it was shown that benzyl isothiocyanates readily bind to proteins (Petri et al., 2020), creating conjugates that will be undetectable in the GC/MS. We did measure and estimate the decrease in BenzylGSL by measuring its intake and its content in the droppings (Fig. 2d). For both naïve and experienced birds' reduction in BenzylGSL was dramatic and similar, suggesting both had similar exposure to BITC.

Method description removed from the results section, and short description added to figures legend to facilitate clarity.

I suggest you use the word naive (vs experienced) rather than unaccustomed - Corrected

I suggest "experienced" and define that the assumption (unless you have tested this and then provide evidence) is that geo-located birds have ecological experience consuming the fruits – Corrected

Clarify if these were isolated from the overlap and non-overlap birds and if not, from what species. Is it possible that BITC only inhibits these enzymes isolated from other species. Corrected

Provide rationale for this prediction (the importance of amylase and lipase) – reference added

I suggest you introduce the natural history earlier through a study system section that helps establish rationale for hypotheses - corrected

If you have any papers that confirm diet by these populations, include them – reference added

Is there variation in this enzyme among species, if not provide that rationale for why this assay represents what might happen in your bird species. include any evidence that these are likely to be similar to wild birds or caveat that your wild birds may have resistance to BITC – which did occur with terpenes and grouse that eat terpenes compared to chickens. See Kohl et al. 2015.

At least include this as potential additional mechanisms for tolerance to BITC by overlap birds.- This is always an issue, but to test this option we would have to slaughter many birds before and after eating *O. baccatus* fruits, extract the enzymes from their intestine and measure activity (as done in Kohl et al. 2015). This is doable but not desirable and we will rather avoid the process. We also feel that the tight correlation between the timing of shift in *Pantoea* population and digestion ability together with the effect of BITC on commercial putrefied amylase and its removal by the bacteria strongly support the suggested theory – i.e. that the *Pantoea* bacteria consume BITC and thus remove its effect on the digestive enzymes – without the need for more complex explanations, more suitable for specialized feeders (e.g. as described in Kohl et al. 2015). We did add text explaining the logic behind the use of commercial enzymes, and the possible caveats.

At what interval and stored how (regrinding droppings samples used for DNA extraction) – corrected

It is possible that the two species eat different amounts of BenzylGSL because they are selective relative to what was given, have host and microbial capacity to reduce hydrolysis or neutralize or metabolize the products. it seems like the digestive exposure of both the parent compound and the hydrolyzed product could be critical info to help interpret and predict physiological consequence that differ between overlap and not in Fig 1c. – we are not sure what "two species" the reviewer is refereeing to. All birds tested here are of single species,

I did not have any comments on the supplemental documents.

I look forward to reading this paper once published.

Many thanks!

Reviewer #2 (Remarks to the Author):

This excellent and original manuscript falls within the scope of Nature Communication and presents well-designed experiments and analyses. This manuscript focuses on presenting one of the first multi-kingdom dispersal interactions. This manuscript can attract great interest in the field of dispersal ecology. Overall, the manuscript fits well with the journal's standards. I have only a few comments for the authors. Please see them below. Thank you for your positive comments!

Overall manuscript: Please check the whole manuscript for spelling and for odd sentences.

Line 16: Do you mean *Ochradenus baccatus*? - Corrected

Line 31-33: Please rephrase this odd sentence - The sentence was rephrased.

Line 41: „plant fitness and resulting” the "and" is not necessary here. - Corrected

Line 49: do you mean food intake? - Corrected

Line 57: Please rephrase this sentence - The sentence has been rephrased.

Line64: I would rather use the word destroy instead of "degrade" - Corrected

line 81: „overlapping” do you mean overlapping? - Corrected

Line 82: missing „i” from distribution - Corrected

Line 87: „overlapping” here again... - Corrected

Line 88: correct the word influence - Corrected

Line 92: missing c from acclimated - Corrected

Methods:

How long the fruits were stored? - The fruits were collected fresh once every three days in the field and stored at 4°C. This is now stated in the method, titled study species

Reviewer #3 (Remarks to the Author):

In this comprehensive study, authors unravel a set of concatenated interactions among a plant, a bird and bacteria and their effects on the participant surrogate of fitness. In such a tripartite cross-kingdom interaction, using a series of well-designed experiments and measurements, authors reveal that the interaction between the plant and its avian seed disperser is critically mediated by *Pantoea* bacteria that reside in both the plant and the bird's intestine. I certainly believe that this manuscript is novel by significantly contributing to open new avenue of research on how microbes mediate plant-animal interactions; however, I miss in the text a mention to a few previous studies that are relevant in such research context (see below).

Thank you for your encouraging review!

Specific comments:

L13. ...in their droppings and regurgitates. - Corrected

L135. Interestingly, ... - Corrected

L222-224, L244-245. Authors argue that they demonstrate for the first time that a specific bacterial species can mediate a multitrophic interaction positively affecting both the plant and the birds. In a previous study, Peris and collaborators (2017) found that vertebrate frugivores, fleshy-fruited plants, and microbes may form a tripartite cross-kingdom interaction in which each part interact positively with the other two. Specifically, they propose that orange (*Citrus sinensis*) seeds and *Penicillium digitatum* spores are both dispersed by mammals, and that fungal infection of orange fruits facilitates their encounter and exploitation by frugivores (mammals and birds). The seminal work by Janzen (1977) is also relevant in the context of plant-seed disperser interactions mediated by microbes. – We wish to thank the reviewer for drawing our attention to these references. The sentence was corrected and a reference to Peris et al., 2017 added.